# Hydrogen Sulfide (H_2_S)/Polysulfides (H_2_S_n_) Signalling and TRPA1 Channels Modification on Sulfur Metabolism

**DOI:** 10.3390/biom14010129

**Published:** 2024-01-19

**Authors:** Hideo Kimura

**Affiliations:** Department of Pharmacology, Faculty of Pharmaceutical Sciences, Sanyo-Onoda City University, 1-1-1 Daigaku-Dori, Sanyo-Onoda 756-0884, Yamaguchi, Japan; kimura@rs.socu.ac.jp; Tel.: +81-836-39-9128

**Keywords:** hydrogen sulfide, polysulfides, neurotransmitter release, S-sulfuration, hippocampal LTP, 3MST, TRPA1 channel

## Abstract

Hydrogen sulfide (H_2_S) and polysulfides (H_2_S_n_, n ≥ 2) produced by enzymes play a role as signalling molecules regulating neurotransmission, vascular tone, cytoprotection, inflammation, oxygen sensing, and energy formation. H_2_S_n_, which have additional sulfur atoms to H_2_S, and other S-sulfurated molecules such as cysteine persulfide and S-sulfurated cysteine residues of proteins, are produced by enzymes including 3-mercaptopyruvate sulfurtransferase (3MST). H_2_S_n_ are also generated by the chemical interaction of H_2_S with NO, or to a lesser extent with H_2_O_2_. S-sulfuration (S-sulfhydration) has been proposed as a mode of action of H_2_S and H_2_S_n_ to regulate the activity of target molecules. Recently, we found that H_2_S/H_2_S_2_ regulate the release of neurotransmitters, such as GABA, glutamate, and D-serine, a co-agonist of N-methyl-D-aspartate (NMDA) receptors. H_2_S facilitates the induction of hippocampal long-term potentiation, a synaptic model of memory formation, by enhancing the activity of NMDA receptors, while H_2_S_2_ achieves this by activating transient receptor potential ankyrin 1 (TRPA1) channels in astrocytes, potentially leading to the activation of nearby neurons. The recent findings show the other aspects of TRPA1 channels—that is, the regulation of the levels of sulfur-containing molecules and their metabolizing enzymes. Disturbance of the signalling by H_2_S/H_2_S_n_ has been demonstrated to be involved in various diseases, including cognitive and psychiatric diseases. The physiological and pathophysiological roles of these molecules will be discussed.

## 1. Introduction

Since the discovery of H_2_S as environmental toxic gas by Ramazinni in 1713, many studies have been devoted to its toxicity. Survivors from the exposure to high concentrations of H_2_S suffered from mental dysfunction, suggesting that the brain must be a target of H_2_S. Warenycia et al. discovered a certain amount of H_2_S existed in the brain [1]. This observation suggested that endogenous H_2_S may have a physiological role in mammals.

The toxic effect of high concentrations of H_2_S on the levels of neurotransmitters has been studied. Exposure of rats to high levels of H_2_S during perinatal development significantly decreases neurotransmitter levels [2]. Sublethal doses of NaHS chronically applied to rodents suppressed neurotransmitter levels [3]. However, the effect of endogenous levels of H_2_S on neurotransmitters release has not yet been well understood.

Cystathionine β-synthase (CBS) and cystathionine γ-lyase (CSE), which catalyse a pathway to produce cysteine, also play a role in producing H_2_S [4,5,6]. Although CSE levels are very low in the brain, it plays an important role under pathological conditions such as Alzheimer’s disease and Huntington’s disease [7,8,9]. 3MST produces not only H_2_S but also polysulfides, including H_2_S_n_ [10,11,12]. Cysteine aminotransferase (CAT) metabolizes L-cysteine and α-ketoglutarate to produce 3-mercaptopyruvate (3MP), which is the substrate of 3MST. 3MP is also produced from D-cysteine by D-amino acid oxydase [13]. The high levels of D-cysteine recently identified in embryonic brain suggest that this pathway plays an important role especially in the developing brain [13,14]. CBS is localized to astrocytes, and 3MST to both neurons and glia in the brain [10,15,16].

Several other enzymes can also produce polysulfides, including sulfur quinone oxidoreductase (SQR), myoglobin, neuroglobin, catalase, superoxide dismutase (SOD), myeloperoxidase, and cysteinyl tRNA synthetase (CARS) [17,18,19,20,21,22,23]. The chemical interaction of H_2_S with NO also produces polysulfides that have been predicted from the finding of synergistic effect of H_2_S and NO on vascular relaxation [6,24,25,26,27,28,29,30,31,32].

The first step of the metabolism of H_2_S is the reduction of the cysteine disulfide bond of sulfide quinone oxidoreductase (SQR) in mitochondria [33] (Figure 1a). It is predicted that the reduction of the cysteine disulfide bond may cause conformational changes in target proteins. Dithiothreitol (DTT) enhances the activity of NMDA receptors by reducing the cysteine disulfide bond at the ligand binding domain of the receptors [34]. H_2_S facilitates the induction of hippocampal long-term potentiation (LTP), a synaptic model of memory formation, by enhancing the activity of N-methyl-D-aspartate (NMDA) receptors [5] (Figure 1b).

It has been extensively studied that S-sulfuration (S-sulfhydration), an addition of sulfur to thiol, modified the activity of enzymes, including cystathionine synthase-serine dehydratase, aldehyde oxidase, and adenylate kinase [36,37,38]. S-sulfuration has been proposed as a mode of action of H_2_S [39]. The activity of glyceraldehyde 3-phosphate dehydrogenase (GAPDH) was increased via S-sulfuration by H_2_S [39].

TRP channels are known to be sensitive to various stimuli including heat and pungent substances including H_2_S_n_ to induce Ca^2+^ influx in cells [11]. We, together with others, have recently reported that TRPA1 and TRPV1 are involved in the metabolism of sulfur-containing substances including methionine and cysteine [40,41]. They even regulate the transcription or translation of 3MST, rhodanese (TST), and persulfide dioxygenase or ethylmalonic encephalopathy protein 1 (ETHE1) [41,42]. It is intriguing to clarify the mechanism for the regulation of the sulfur metabolism by the channels.

## 2. Endogenous H_2_S, Its Production, and Its Metabolism

Warenycia et al. detected H_2_S by the methylene blue method, in which the reaction mixture contains high concentrations of acids that make brain homogenates release great amounts of H_2_S from the iron-sulfur cluster attached to enzymes belonging to the respiratory chain, including aconitase [1,43]. For this reason, the endogenous levels of free H_2_S with this method were over-estimated. Two other groups also measured H_2_S levels in the brains of humans and bovine with a similar method, which showed 50–160 μM [44,45].

Since then, several methods were applied to measuring endogenous H_2_S, and the lowest level estimated in the brain was 14 nM [46]. In their method, the tissue was homogenized with phosphate buffer, and released H_2_S from homogenates was measured by gas chromatography. Because free H_2_S is re-absorbed by the tissue homogenates as sulfane sulfur [43], the value obtained with this method was underestimated.

H_2_S and polysulfides have been measured as their monobromobimane (mBB) adducts by HPLC and LC-MS/MS. The labelling reaction with mBB has been performed under alkaline conditions (pH 8.4–9.5), where the exchange between H_2_S and polysulfides shifts more to H_2_S than to polysulfides, resulting in the overestimation of H_2_S levels and the underestimation of polysulfide levels [47]. We derivatized H_2_S and polysulfides with mBB at physiological pH 7.0. Recent evaluation of H_2_S using LC-MS/MS showed 0.030 ± 0.004 μmol/gram protein, which is approximately 3 μM in the brain [48]. Even with this method, both intracellular and extracellular molecules react with each other at homogenization, which causes extracellular cystine to be reduced to cysteine, which reacts with intracellular enzymes to generate H_2_S.

CBS produces H_2_S from L-cysteine and L-homocysteine. The activity of CBS is regulated by S-adenosyl methionine (SAM), NO, and CO [5,49,50,51]. SAM binds to the site located at the carboxy-terminus of CBS to enhance the activity, while NO and CO bind to the heme group located at the amino-terminus to suppress the activity (Figure 2). This regulation has an important role under hypoxic conditions in the brain. Under hypoxia, CO production is decreased, leading to the enhancement of CBS activity in astrocytes surrounding capillaries, which are dilated by produced H_2_S to restore blood flow [52]. Overproduction of H_2_S in hypoxia, in turn, suppresses mitochondrial cytochrome oxidase, leading to damage in the brain; the damage can be decreased depending on the levels of SQR [53]. However, the levels of SQR have been reported to be very low in the brain [54]. To compensate for the low levels of SQR, neuroglobin has been predicted to play a role in H_2_S oxidation in the brain [19].

CBS was reported to be also localized to neurons using the polyclonal antibody preincubated with CBS knockout mouse brain extract to mask any nonspecific immunoreactivity [55]. However, not only our antibody but also the same antibody as mentioned by Robert et al., after affinity purification, detected CBS in astrocytes but not in neurons [15].

CSE produces H_2_S under physiological low concentrations of Ca^2+^, while its activity is suppressed by high concentrations of Ca^2+^ [56]. Once intracellular concentrations of Ca^2+^ are increased in stimulated cells, H_2_S production by CSE can be decreased. The levels of CSE are also transcriptionally regulated. The promoter region of the CSE gene has an SP-1 binding site, and the expression of CSE is enhanced three times by tumour necrosis factor α, which enhances the binding of SP1 [7,57].

3MST produces H_2_S from 3MP, a product of CAT from L-cysteine and α-keto glutarate [10,58,59,60]. The active-site cysteine residue of 3MST receives sulfur from 3MP, then S-sulfurated active-site cysteine residue of 3MST releases H_2_S via reduction by thioredoxin (Figure 3) [61,62]. 3MST is localized to the cytoplasm as well as to mitochondria, in which the level of cysteine is sufficiently high (approximately 1 mM) to produce H_2_S [63,64].

Propargylglycine and β-cyano-L-alanine have been used to inhibit the activity of CSE, and aminooxyacetic acid inhibits CBS [4,66,67,68]. Recently, a specific inhibitor of 3MST has been developed by Hanaoka et al. (2017) by high-throughput screening of a large chemical library (174,118 compounds). Their compound 3 with aromatic ring-carbonyl-S-pyrimidone structure specifically interacts with the persulfur anion of the S-sulfurated cysteine residue of 3MST to suppress the production of H_2_S and polysulfides [69].

## 3. Bound Sulfane Sulfur and S-Sulfurated Proteins

In tissues, there are forms of sulfur, sulfane sulfur, which release H_2_S under reducing conditions [3,43,70]. They include H_2_S_n_, Cys-SSH, GSSH, and P-SSH. Warenycia et al. identified the non-acid labile H_2_S liberated by DTT from brains of H_2_S-poisoned animals [3], and it was later called bound sulfur [70]. The levels as well as the characteristics of bound (sulfane) sulfur are different between tissues. Homogenates of the liver released H_2_S, which was increased and reached its maximum at one hour (approximately 6 μmol H_2_S/g protein) and then decreased abruptly after 2 h of DTT application. In contrast, those of the brain gradually increased H_2_S release up to 5 h after DTT addition (approximately 2 μmol H_2_S/g protein) [43]. Homogenates of the heart released H_2_S just less than approximately 0.2 μmol H_2_S/g protein for the first two hours, then stopped releasing.

The absorption rates of H_2_S are also different between tissues. Liver homogenates promptly absorbed H_2_S, while those of hearts took 10 min to completely absorb it and those of brains took 30 min [43]. The interesting thing to note is that heart homogenates released a little H_2_S by DTT, while it absorbed H_2_S well, suggesting that cysteine residues may be highly oxidized in this tissue, or that the abundant iron reacts with H_2_S. These observations indicate that each tissue has the specific forms of sulfane sulfurs, which have the characteristics of a different rate of H_2_S release and absorption, probably due to having the different redox states.

## 4. Production of H_2_S_n_ and Other S-Sulfurated Molecules

Endogenous H_2_S_n_ have been identified in the brain [11,48]. 3MST produces H_2_S_n_ and other S-sulfurated molecules, such as cysteine persulfide (Cys-SSH), glutathione persulfide (GSSH), and S-sulfurated cysteine residues of proteins (P-SSH) [12,71]. Two possible mechanisms have been proposed. Sulfur is transferred from S-sulfurated active cysteine residues of 3MST to the acceptor molecules such as H_2_S, cysteine, glutathione, and cysteine residues, to produce corresponding S-sulfurated molecules (Figure 4a) [12,71]. Alternatively, 3MST produces H_2_S_2_, which reacts with cysteine, glutathione, and cysteine residues to generate their S-sulfurated molecules (Figure 4b) [12,71]. In the latter mechanism, H_2_S_2_ has an advantage in easily accessing targets with less steric hindrance comparing with 3MST itself, but less specific to the target molecules. Recently, 3MST was shown to directly transfer sulfur to cysteine residues using 3MST fused with redox-sensitive green fluorescent protein, which has two thiol groups on its surface and acts as a highly efficient sulfur acceptor from 3MST [72].

Two mechanisms have been proposed for S-nitrosylation of proteins by NO (Figure 5). NO is defused to the target proteins to S-nitrosylate them (Figure 5a) [74]. Another mechanism is that NO-synthase makes a complex with S-nitrosylase to S-nitrosylate their target proteins (Figure 5b) [75]. Considering these mechanisms for S-nitrosylation, S-sulfuration may also be mediated, both by the diffusion of small sulfur molecules such as H_2_S and H_2_S_n_ as well as by enzymes including 3MST.

3MST is also known as tRNA thiouridine modification protein [76,77]. 3MST transfers sulfur from 3MP to uridine of tRNA that provides accurate deciphering of the genetic code and stabilizes the structure of tRNA [77,78,79]. Cysteinyl tRNA synthetase (CARS) has been found to transfer sulfur from one cysteine to another to produce Cys-SSH [23].

The intracellular concentrations of H_2_S are well regulated by a balance of its production and clearance [80,81]. SQR receives sulfur from H_2_S to its active-site cysteine to produce persulfide (Figure 1a), whose sulfur, in turn, has been proposed to be transferred to H_2_S, cysteine, and GSH to produce H_2_S_2_, Cys-SSH, and GSSH, respectively [17]. Haemoglobin and myoglobin localized to blood and muscle, respectively, oxidize H_2_S to H_2_S_n_ [18,81]. Neuroglobin, another haemoprotein located in neurons, also oxidizes H_2_S to H_2_S_n_ [19].

Catalase, which is highly S-sulfurated and may be regulated by H_2_S, H_2_S_n_, and other S-sulfurated molecules [39], oxidizes H_2_S to produce H_2_S_n_ [20]. Copper/zinc super oxide dismutase (SOD) generates H_2_S_n_ by oxidizing dissolved H_2_S gas rather than HS^−^ in the presence of O_2_ and H_2_O_2_ [21,82,83]. Mitochondrial manganese SOD can also oxidize H_2_S to H_2_S_n_. Peroxidases such as lactoperoxidase and myeloperoxidase oxidize H_2_S to H_2_S_n_ in the presence of H_2_O_2_ [22,84].

CSE has been proposed to be able to produce Cys-SSH from cystine [85], and CBS was recently added to a group of enzymes to have a similar activity to CSE [86]. Both CSE and CBS are localized to cytoplasm, where cystine levels are approximately 0.20 and 0.05 μM in the liver and lung, respectively, and under detectable levels in the brain and heart, while the Km value of CSE to cystine is 30 to 70 μM [86,87]. If the intracellular cystine concentrations are above 5 μM, CSE may have a possibility to synthesize Cys-SSH in cells [88]. It was recently reported by the same group that the production of H_2_S, H_2_S_n_, Cys-SSH, and GSSH was not changed in mice with triple knock-out of CBS, CSE, and 3MST [89].

## 5. H_2_S_n_ Production by the Chemical Interaction between H_2_S and NO

H_2_S relaxes vascular smooth muscle in synergy with NO [6], and a similar synergistic effect was observed in intestine [90]. H_2_S_2_ and H_2_S_3_ are produced from H_2_S and NO [31,32]. H_2_S_n_ exist in the brain as products of oxidized H_2_S as well as those generated by 3MST [11,12,48].

As a mechanism for the generation of H_2_S_2_ from H_2_S and NO, NO is oxidized to N_2_O_3_, which reacts with H_2_S to generate HSNO. HSNO, in turn, reacts with H_2_S to produce H_2_S_2_ [29,91].
2NO + O_2_ = 2NO_2_
NO_2_ + NO = N_2_O_3_
N_2_O_3_ + H_2_S = HSNO + H + NO_2_^−^
HSNO + H_2_S = HSSH + HNO

Alternatively, one electron oxidation of HS^−^ generates thiyl radical (HS^•^), which readily react with NO radical to produce HSNO.
HS^−^ = HS^•^ + e^−^
HS^•^ + NO^•^ = HSNO

The production of H_2_S_n_ has also been demonstrated in peritoneal mast cells, where the application of H_2_S donor GYY4137 [92] and NO donor DEA NONOate reacted with endogenous NO and H_2_S, respectively, in the cells, and the products were detected by a polysulfide-sensitive fluorescent probe SSP4 [31,93]. The potential products, H_2_S_n_, were detected by the activation of Ca^2+^ channels in mast cells.

Ebenhardt et al. proposed that nitroxyl (HNO) together with H_2_S_n_ are produced by the interaction of H_2_S with NO, and that HNO is a major product which activates TRPA1 channels [29]. The product from the interaction between H_2_S and NO as well as H_2_S_n_ are degraded by cyanide, while HNO is resistant, suggesting that H_2_S_n_ may be a preferable chemical entity to HNO to activate TRPA1 channels [32]. Cortese-Krott et al. reported the generation of SSNO^−^, H_2_S_n_, and dinitrososulfite from the interaction of H_2_S with NO [30]. The products from H_2_S and NO as well as H_2_S_n_ are degraded by reduction, while SSNO^−^ is resistant to reduction [30,32]. Based on these observations, we suggested that H_2_S_n_ are the potential products [32]. Recently, Bogdandi et al. reported that H_2_S_n_ is less stable compared to SSNO^−^ and suggested that H_2_S_n_ transiently activate TRPA1 channels at the early phase of the production of these substances, while more stable SSNO^−^ induces sustained activation of the channels [94].

## 6. S-Sulfuration of Cysteine Residues by H_2_S_n_

Sulfur atoms with the same oxidation state are not able to react with each other. Because the oxidation state of both sulfur atoms in H_2_S and that in cysteine is −2, they do not react with one another. In contrast, in H_2_S_n_, the oxidation state of sulfur atoms is 0 or −1. TRPA1 channels were found to be activated by H_2_S, and it was later identified that H_2_S_n_ activate the channels more efficiently than H_2_S [11,95,96,97]. TRPA1 channels have 31 cysteine residues, and 21 are localized to the amino terminus [98]. Cys422 and Cys634 are sensitive to H_2_S_n_ to be S-sulfurated [99]. One of the two cysteine residues may be S-sulfurated and reacts with another to make a disulfide bond to induce the conformational changes to activate the channels (Figure 6) [12]. Alternatively, two cysteine residues are S-sulfurated at the same time. The former may more efficiently change the conformation of the channels. The activity of tumour suppressor PTEN is modified by H_2_S_n_ through two cysteine residues, Cys71 and Cys124, forming a disulfide bond [100]. A monomer protein kinase G1α (PKG1α) is inactive, while the formation of a cysteine disulfide bond between each Cys42 through S-sulfuration by H_2_S_n_ makes the active homodimer (Figure 6) [101].

ATP-dependent K^+^ channels are activated through S-sulfuration of Cys43 in its Kir subunits by H_2_S [102]. Cys43 is located at the critical site for the binding to ATP, and the subtle conformational change by S-sulfuration may cause the dramatic changes in binding of the channels to ATP. Cys43 is exposed to the cytoplasmic side of cells, and if it is Cys-SH form, H_2_S_n_ rather than H_2_S S-sulfurate it. In contrast, if Cys43 is in Cys-SNO or Cys-SOH form, H_2_S is preferable to H_2_S_n_ to S-sulfurate it. The other example is GAPDH [39,103]. The activity of GAPDH was initially reported to be increased via S-sulfuration by H_2_S [39]. However, it was later reported that H_2_S had no effect on GAPDH activity, while H_2_S_n_ inhibited GAPDH [103]. The possible mechanism is discussed in the next section.

## 7. S-Sulfuration of S-Nitrosylated or S-Sulfenylated Proteins by H_2_S

The human cysteine proteome includes 214,000 cysteine residues and 5–12% of total cysteine residues are oxidized, including the cysteine disulfide bond (P-S-S-P), P-SNO, P-SOH, and P-SSH [104]. It has been predicted from the fact that tissues contain bound (sulfane) sulfur [3,43,70,105]. Mustafa et al. identified 39 S-sulfurated proteins, including GAPDH, β-tubulin, and actin, in liver lysates [39].

The oxidation state of sulfur in these oxidized cysteine residues is −1 or 0, and they are able to react with H_2_S. P-S-S-P, P-SNO, and P-SOH cysteine residues are reduced by H_2_S to P-SSH, which is further reduced to P-SH. When H_2_S and NO are generated at a restricted area in cells, either one of two reactions may occur: H_2_S_2_ is produced from H_2_S and NO, and then S-sulfurate P-SH to P-SSH. Alternatively, NO reacts with cysteine residues to generate P-SNO, which is reduced by H_2_S to P-SSH.

Some cysteine residues may have specificity to the reaction with NO, while others react with little preference. For example, among around 100 cysteine residues in the skeletal muscle ryanodine receptor, many of them are oxidized, but only one of them is S-nitrosylated [106]. In parkin, of which the mutations cause Parkinson’s disease, Cys95, Cys59, and Cys 182 are S-sulfurated in the brains of normal individuals, while they are S-nitrosylated in those of patients [107]. Two cysteine residues of the amino terminus of TRPA1, Cys 422, and Cys634 in mouse are responsible for activation by H_2_S_2_ and H_2_S_3_ [11,32,99], while the human equivalence Cys421 and Cys633 have been reported to be sensitive to O_2_ [108]. Cys 421 in human is also modified by NO and H_2_O_2_ [109].

H_2_O_2_ reacts very slowly with free cysteine (2–20 M^−1^·s^−1^). In contrast, its reactivity significantly increases to cysteine residues (10–10^6^ M^−1^·s^−1^) [110]. Although H_2_O_2_ poorly reacts with H_2_S (0.73 M^−1^·s^−1^), P-SOH react two orders of magnitude faster with H_2_S than with glutathione at pH 7.4 [111,112]. The activation of receptor tyrosine kinase enhances the production of H_2_O_2_ and the subsequent S-sulfenylation of target proteins [113,114]. Zivanovic et al. demonstrated that the application of epidermal growth factor (EGF) to cells induced dramatic increase in the levels of P-SOH, and then they decreased [115]. At the decreasing phase of P-SOH, the levels of P-SSH started increasing. This correlates well with the increase in the expression of CBS, CSE, and 3MST, suggesting that P-SOH efficiently reacts with H_2_S to produce P-SSH.

## 8. H_2_S and H_2_S_n_ in Neuronal Transmission

The other gaseous signalling molecules, NO and CO, are recognized as retrograde neurotransmitters, which are released from postsynapse to presynapse to facilitate the release of a neurotransmitter glutamate, to induce hippocampal LTP [116,117,118,119,120,121]. When postsynaptic NMDA receptors are activated by glutamate, Ca^2+^ enters cells through NMDA receptors and makes Ca^2+^/calmodulin complex, which activate NO synthetase (NOS) to release NO. NO diffuses to the presynapse, where it activates soluble guanylyl cyclase to produce cGMP, which activates G-kinase to modulate the release of neurotransmitter glutamate, leading to facilitating LTP induction [122,123].

As described previously, the enhancement of NMDA receptor activity through its reduction was not the only mechanism for the facilitation of LTP induction by H_2_S. Although the reducing activity of H_2_S is weaker than that of DTT, H_2_S more efficiently facilitated the induction of LTP than DTT [5], suggesting an additional mechanism for facilitating LTP induction.

We found that H_2_S activates astrocytes to induce Ca^2+^ influx via the TRP family of channels [96], and Streng et al. identified that TRPA1 channels in the bladder are activated by H_2_S [97]. H_2_S_n_ have much higher potency than H_2_S to induce Ca^2+^ influx by activating TRPA1 channels in astrocytes [11,95,96,97,124,125]. Endogenous H_2_S_2_, H_2_S_3_, cysteine per- and poly-sulfide, and glutathione per- and poly-sulfide were identified in the brain [11,12,48,71,126], and the garlic-derived organic polysulfides dimethyl trisulfide (DMTS) and diallyl trisulfide (DATS) also activate TRPA1 channels [11]. Shigetomi et al. reported that the activation of TRPA1 channels induced the release of D-serine, a coagonist of NMDA receptors, from astrocytes to facilitate the induction of LTP [127]. Based on these observations, the additional mechanism for LTP induction was predicted to be the activation of TRPA1 channels in astrocytes by H_2_S_n_ to induce a release of D-serine [11,127,128]. Recently, however, D-serine was reported to be mainly released from neurons, to which TRPA1 channels are not localized, and that the contribution of astrocytes to release D-serine is minor [129]. It was necessary to re-evaluate the involvement of TRPA1 channels in the release of D-serine from astrocytes. Recent findings are discussed in the next section.

## 9. The Release of Neurotransmitters by H_2_S and H_2_S_n_

Granule cells produce NO and the synaptic terminals contacted with them were thought to be the target of NO. NO induces the release of glutamate, and it has been suggested that cGMP-dependent protein kinase may be involved in the alteration of synaptic vesicle dynamics [130,131].

Mitrukhina et al. reported that NaHS increased the frequency of miniature end-plate potentials and the amplitude of the evoked postsynaptic responses to a single stimulation in the motor nerve ending of the mouse diaphragm, suggesting the enhanced synaptic vesicle exocytosis and reduced endocytosis [132]. The same group showed that a similar observation was obtained in the neuromuscular junction showing the involvement of intracellular Ca^2+^, cAMP, and presynaptic ryanodine receptors in the vesicular release of neurotransmitters [133].

We recently found that H_2_S and H_2_S_2_ induce the release of neurotransmitters, GABA, glutamate, glycine, and that of neuromodulators, D-serine, L-serine, and glutamine using in vivo microdialysis (Figure 7) [41]. GABA and glutamate were greatly released by H_2_S and to a lesser extent by H_2_S_2_, while D-serine, L-serine, and glutamine were released less efficiently than GABA and glutamate, probably because their release mechanisms may be different. Considering the observations that H_2_S more greatly induced the release of GABA and glutamate than H_2_S_2_ and that the endogenous levels of H_2_S are greater than those of H_2_S_2_, H_2_S rather than H_2_S_2_ may mainly regulate the release of neurotransmitters [41]. In the in vivo microdialysis, the applied concentrations of Na_2_S and Na_2_S_2_ are dramatically decreased when they are released from the dialysis filter probe to the brain. The in vitro recovery from microdialysis probes has been reported between 0.1% and 7.6%, depending on molecules [134], and our study showed that Na_2_S was recovered approximately 0.2–1% and Na_2_S_2_ 0.8–4.8% [41]. Based on these observations, a concentration of Na_2_S and Na_2_S_2_ a minimum of 50 times higher (1 mM) than that used for the brain cell suspension (20 μM) was applied.

The release of D-serine by H_2_S_2_ from the hippocampus of TRPA1 knockout rats showed no significant difference with that in the wild-type rats [41]. However, LTP was not induced in TRPA1 knockout rats, which agrees with the observation by Shigetomi et al. [127]. The synaptic transmission is regulated by the interaction between neurons and astrocytes, and astrocytes-dependent facilitation regulates the vesicular release of neurotransmitters including glutamate in the hippocampal synapses (Figure 8) [135]. These observations suggest that the activation of TRPA1 channels in astrocytes is required for the induction of LTP, but that the channels are not involved in the release of D-serine [41].

In the brain cell suspension, serotonin suppressed the release of H_2_S, while it increased that of H_2_S_3_ [41]. Acetylcholine increased the intracellular levels of H_2_S and the release of cysteine persulfide, and norepinephrine and GABA increased the release of cysteine persulfide [41]. These observations suggest that H_2_S and H_2_S_2_ facilitate the release of neurotransmitters, while neurotransmitters in turn regulate the release of H_2_S and H_2_S_2_. Probably due to the low recovery of H_2_S and H_2_S_2_ through the probe of in vivo microdialysis, the release of both molecules induced by the neurotransmitters was not detected [41].

Since the discovery of H_2_S_n_ activating TRPA1 channels, PTEN, and protein kinase G1α, the effects of H_2_S previously found were thought to be mediated by H_2_S_n_ [11,100,101]. However, H_2_S releases neurotransmitters more efficiently than H_2_S_2_ as described above. Moreover, H_2_S and H_2_S_2_ exert with a similar potency the neuroprotective effects against oxidative stress by increasing the intracellular concentrations of glutathione, the major endogenous antioxidant, through the activation of cystine/glutamate antiporter and cysteine transporter [136]. Alanine-serine-cysteine transporter (Asc-1), a neutral amino acid transporter located in the neurons, and Slc38a1 localized to GABAergic neurons in the hippocampus are candidate molecules releasing D-serine, L-serine, and glutamine [137,138,139]. These transporters may also be involved in the regulation of the release of GABA, glutamate, and glycine induced by H_2_S and H_2_S_2_.

The cytoprotective effect of H_2_S has been observed in other organs. H_2_S protects cardiac muscle by preserving the function of mitochondria [140], and mitochondrial H_2_S produced by 3MST regulates the catabolism of branched-chain amino acid to protect cardiac muscle from heart failure [141]. H_2_S-based therapy has the potential to attenuate the progression of Duchenne muscular dystrophy and other muscle-related diseases [142,143,144,145].

## 10. TRPA1 Channels Are Involved in Sulfur Metabolism

TRP channels are not only the target of H_2_S_n_, but their regulation on sulfur metabolism and the levels of neurotransmitters have recently been reported (Figure 9). TRPA1-suppressed (homozygotes)-Drosophila showed decreased levels of methionine and cysteine, 15 and 50%, respectively, of the wild type [40]. The application of resiniferatoxin, an ultrapotent capsaicin analogue antagonist of TRP vanilloid type 1 (TRPV1) and TRPA1 channels [42,146], to porcine urinary bladder increased the transcription of rhodanese, and decreased that of sulfur dioxygenase (ETHE1) and molybdenum cofactor synthesis 2, which are involved in the metabolism of sulfur [42].

We found that the levels of 3MST and those of cysteine were lower in TRPA1 knockout rats compared to those in the wild type, while those of rhodanese (TST) were not changed [41]. The endogenous levels of H_2_S, H_2_S_2_, and cysteine persulfide in the brain of TRPA1-KO rats are also lower than those in the wild type, probably due to the low levels of both 3MST and cysteine, which is a substrate of CAT to produce 3MP [41].

TRPA1 channels also may regulate the levels of neurotransmitters. Deficiency of TRPA1 channels decreases the levels of glycine, while increases those of GABA [40,41]. The application of resiniferatoxin upregulates spermidine/spermine N(1)-acetyltransferase 2 (SAT2), which regulate the uptake of glutamate at the synaptic terminal, while suppresses the regulator of G protein signalling 7 (RGS7), which regulates synaptic plasticity and GABA signalling [42]. These observations suggest that TRPA1 channels not only respond to H_2_S_n_, but also regulate the production of H_2_S/H_2_S_2_ and the neurotransmission by glutamate and GABA.

The activation of TRPA1 channels induces Ca^2+^ influx to increase the intracellular concentrations of Ca^2+^ that suppresses the activity of CAT [143]. In the absence of increase in the intracellular Ca^2+^ though TRPA1 channels, CAT may uncontrollably metabolize cysteine to produce excess 3MP [40,143], which may induce feedback inhibition on the production and transport of cysteine. It may cause a decrease in the transport of cysteine from astrocytes to neurons. The transcription or translation of sulfur-metabolizing enzymes including 3MST may also be regulated by TRPA1 channels [41,42]. Further studies are awaited to understand this mechanism.

## 11. Pathophysiological Roles of H_2_S and H_2_S_n_

High concentrations of H_2_S are toxic, while low concentrations have benefits. Down’s syndrome (DS), characterized by impaired brain growth and maturation that causes mental retardation, has a trisomy of chromosome 21 where CBS is encoded. CBS in the brains of patients is approximately three times more abundant compared to that of normal individuals [147]. The urine of patients contains high levels of thiosulfate, a metabolite of H_2_S [148]. The overexpression of CBS showed DS like neurocognitive deficit in mice [149]. Fibroblasts prepared from DS patients showed suppression in the mitochondrial electron transport, oxygen consumption, and ATP generation [150].

Ethylmalonyl encephalopathy is an autosomal recessive early onset and defective in cytochrome c oxidase in the brain and muscle. In this disease, sulfur dioxygenase is deficient, resulting in increasing H_2_S levels that suppress cytochrome c oxidase [151].

Both excess and deficiency of H_2_S and H_2_S_n_ have been proposed to be involved in the pathogenesis of schizophrenia. Plasma H_2_S levels, which have a correlation with working memory and executive function, are significantly lower in patients than those of normal individuals [152]. Thiol homeostasis is shifted to more oxidized disulfide bond formation from free thiol in patients [153]. In contrast, it has been reported that a mouse model of schizophrenia showed greater levels of bound sulfur than a control. The levels of CBS, CSE, and 3MST in post-mortem brain of schizophrenia patients are significantly greater than those of normal individuals [154].

Hypersensitivity to NMDA receptor antagonists including MK-801 resembles positive symptoms in schizophrenia [155]. Although MK-801 induced hyperlocomotion even in the wild-type rats, a significant increase was observed in the locomotor activity after MK-801 administration in 3MST-KO rats compared to the wild-type rats [41]. Working memory examined with Y-maze showed that the total arm entry of 3MST-KO was significantly less than that of the wild type, but the percent spontaneous alternation was not significantly different between 3MST-KO and the wild-type rats. The other behavioural tests, the open field, PPI, and contextual fear conditioning tests did not show any significant difference between the genotypes in the rat model. The difference in PPI between mice and rats models may be involved, probably due to the increase in the levels of rhodanese in mice, while no changes in rats [41,156].

Gliomas with the highest grades of malignancy contained the greater levels of polysulfides than glioma-free brain regions [157]. A similar observation has been reported that H_2_S_n_ levels were greater in glioblastoma-bearing regions than glioblastoma-free control regions [158]. In contrast, CBS-silenced glioma increased the levels of VEGF and HIF-2α, deeply invaded with dense vascularization, and aggressively proliferated [159]. Further studies are awaited to clarify the involvement of H_2_S and its producing enzymes in cancer pathology.

## 12. Perspective

Since the identification of H_2_S_n_ regulating the activity of TRPA1 channels, PTEN, and protein kinase G1α [11,100,101], the effects of H_2_S previously reported had been thought to be mediated by H_2_S_n_. However, there appeared cases in which H_2_S showed greater effects than H_2_S_n_. H_2_S 10 times more potently induced the release of GABA than H_2_S_2_ in in vivo microdialysis from the hippocampus [41]. Another example is that H_2_S and H_2_S_n_ exerted the cytoprotective effect in a similar fashion on neurons against oxidative stress induced by high concentrations of glutamate [136]. Considering the fact that the endogenous levels of H_2_S are greater than those of H_2_S_n_, H_2_S is responsible for these effects [41]. In these cases, the oxidized cysteine residues of the target proteins are S-sulfurated by H_2_S to generate P-SSH, and even reduced further, probably together with GSH to P-SH. Either H_2_S or H_2_S_n_ transmit signals depending on the form of the target cysteine residues.

The existence of the cysteine residues with sulfur atoms with different oxidation states in proteins has been predicted from the release and absorption of H_2_S from bound sulfane sulfur in tissues. The liver contains more bound sulfane sulfur than the brain. Although the heart has a little capacity to release H_2_S by DTT, it absorbed exogenously applied H_2_S as greatly as the liver, and more effectively than the brain, as described previously [43]. The species and amount of bound sulfane sulfur are different between tissues. It is possible that P-SOH and P-SNO, which do not release H_2_S under reducing condition but efficiently absorb H_2_S, are more abundant in the heart than the other tissues. The balance of cysteine residues with thiols or oxidized forms may cause differences in the transduction of signalling by H_2_S or H_2_S_n_.

In addition to SAM, NO, CO, and Ca^2+^, which regulate the activity of H_2_S- and H_2_S_n_- producing enzymes [5,50,56,61,143,160], neurotransmitters such as serotonin, acetylcholine, norepinephrine, and GABA regulate the release of H_2_S, H_2_S_n_, and cysteine persulfide [48]. The regulation mechanism of H_2_S/H_2_S_n_ production by neurotransmitters, probably by the control of their producing enzymes, are yet to be clarified. The development of fluorescence probes, which are sensitive enough to detect the physiological concentrations of H_2_S, H_2_S_n_, and other S-sulfurated molecules in real time, will clarify the involvement of H_2_S and H_2_S_n_ in regulating neurotransmission in the brain.

Polysulfides including H_2_S_2_ are reactive and unstable, but they still endogenously exist [39,41,43,48,86]. The total redox activity of H_2_S_2_, Cys-SSH, GSSH, and P-SSH is well balanced with that of H_2_S, Cys-SH, GSH, and P-SH in the steady state in cells. It may locally be unbalanced when S-sulfurated molecules are produced, but once signalling has been performed, it returns to the steady state by reacting with the surrounding molecules such as GSH to normalize the redox state. The regulation of the total redox balance may be another problem to be clarified.

The involvement of TRPA1 channels in the metabolism of sulfur-containing molecules and the translation or transcription of sulfur-metabolizing enzymes will open a new field in the function of TRP channels together with that of H_2_S and polysulfides.

## Figures and Tables

**Figure 1 biomolecules-14-00129-f001:**
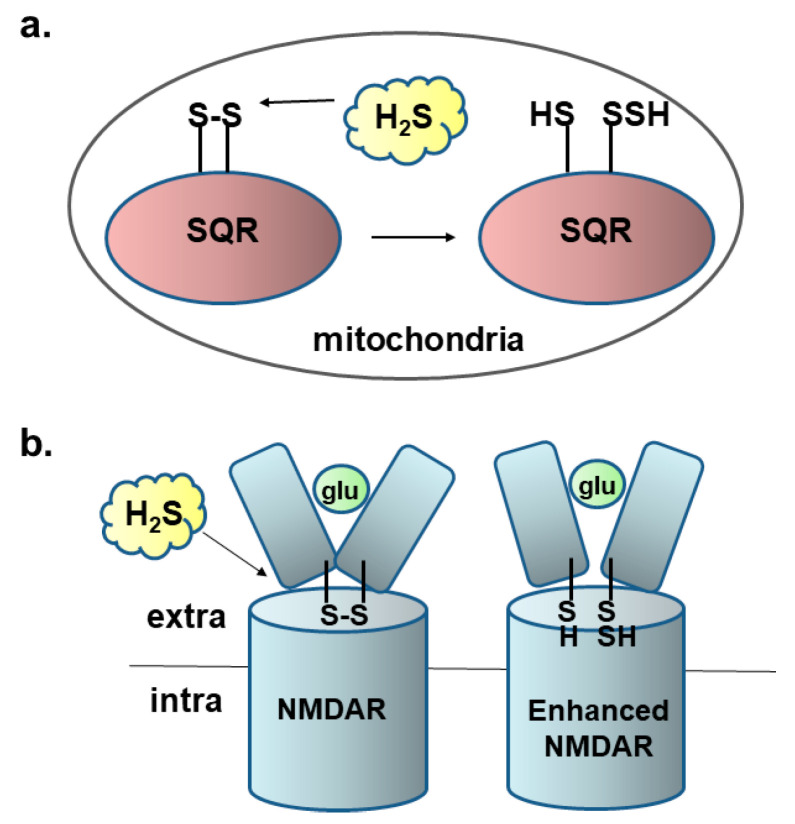
H_2_S reduces the cysteine disulfide bond. (**a**). Sulfur-quinone-oxydoreductase (SQR) metabolizes H_2_S in mitochondria. The first step of the metabolism of H_2_S is the reduction of the cysteine disulfide bond of SQR in mitochondria. (**b**). H_2_S enhances the activity of NMDA receptors. H_2_S reduces the cysteine disulfide bond localized to the hinge of the ligand binding domain of the receptor to enhance the receptors activity. This figure is produced by modifying Kimura (2016) [35].

**Figure 2 biomolecules-14-00129-f002:**
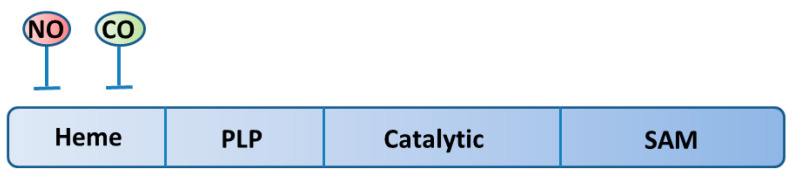
Functional domains of CBS. NO and CO bind to the heme group located at the amino-terminus to suppress the activity, and SAM binds to the site located at the carboxy-terminus of CBS to enhance the activity. NO: nitric oxide, CO: carbon monoxide, PLP: pyridoxal 5′-phosphate, SAM: S-adenosyl-L-methionine.

**Figure 3 biomolecules-14-00129-f003:**
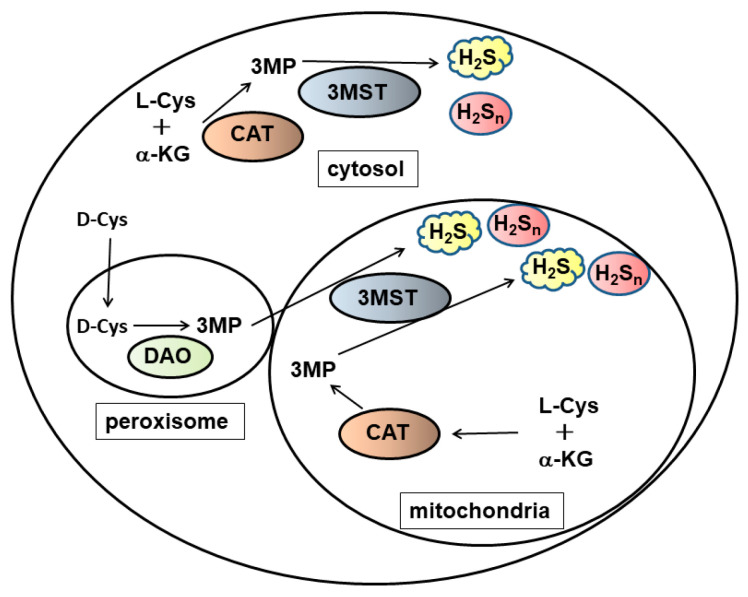
Production of H_2_S and H_2_S_n_ by 3MST. 3MST produces H_2_S and H_2_S_n_ from 3MP, which is produced from L-cysteine and a-ketoglutarate by CAT. This pathway is localized both in the cytoplasm and mitochondria. 3MP is also produced from D-cysteine by DAO in the peroxisome. Peroxisome and mitochondria exchange their metabolites via vesicular trafficking [65]. 3MST: 3-mercaptopyruvate sulfurtransferase, CAT: cysteine amino transferase, DAO: D-amino acid oxidase.

**Figure 4 biomolecules-14-00129-f004:**
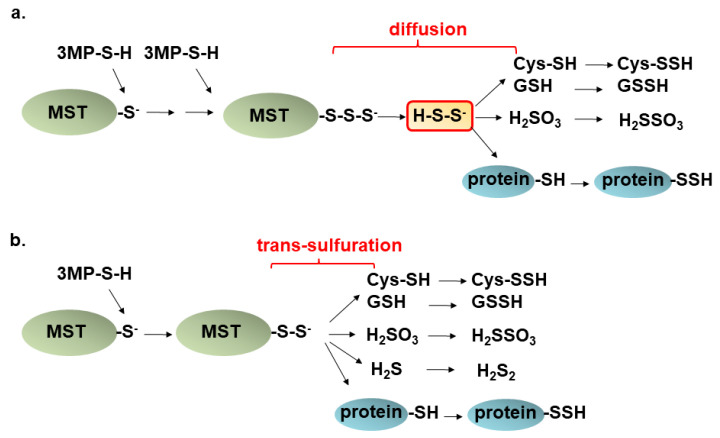
A possible mechanism for the production of S-sulfurated molecules. (**a**). H_2_S_2_ produced by 3MST diffuses to react with molecules containing thiol to produce corresponding S-sulfurated molecules. (**b**). Alternatively, 3MST trans-sulfurates thiols to produce S-sulfurated molecules. This figure is produced by modifying Kimura (2020) [73].

**Figure 5 biomolecules-14-00129-f005:**
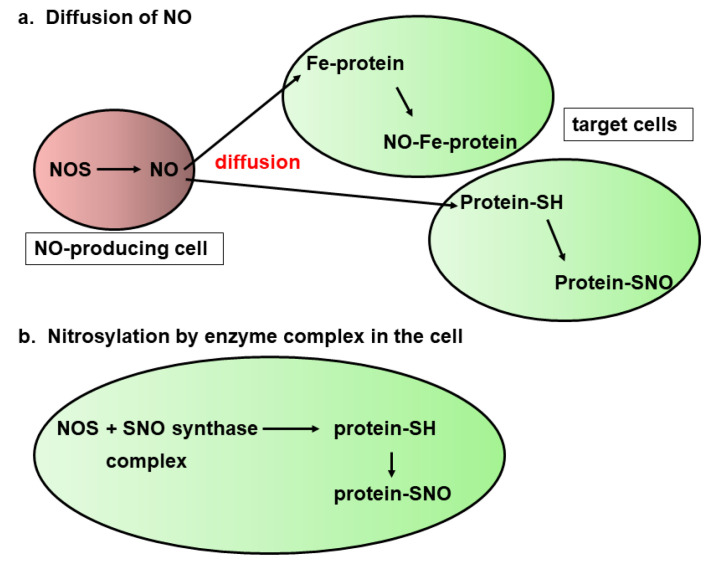
Two pathways proposed for S-nitrosylating the target proteins. After NO is produced by NO synthase, it diffuses to reach the target cysteine residue to S-nitrosylate it, or heme proteins to react with heme. Alternatively, the complex of NO synthase and S-nitrosylase directly S-nitrosylates the target cysteine residues in the cell.

**Figure 6 biomolecules-14-00129-f006:**
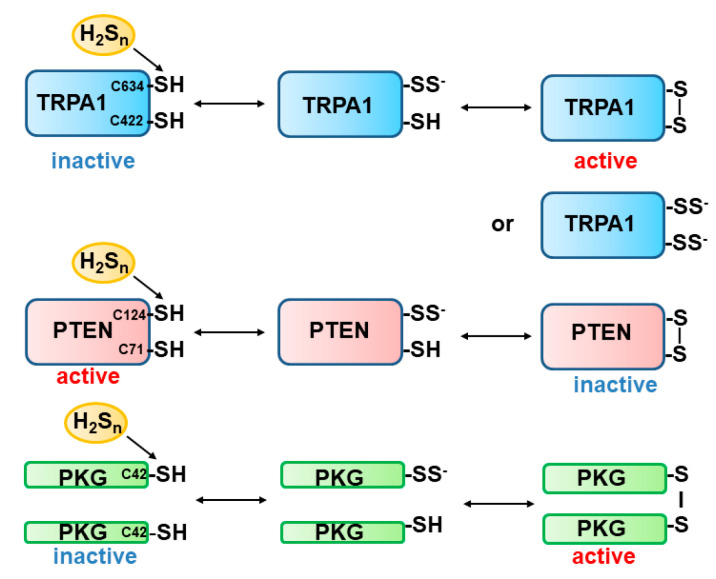
S-sulfuration of cysteine residues of target proteins by H_2_S_n_. TRPA1 channels and PTEN have two cysteine residues sensitive to S-sulfuration to regulate their activity. One cysteine residue is S-sulfurated to react with another thiol to generate a cysteine disulfide bond. PKGα monomer is S-sulfurated to react with another monomer to produce the active-form dimer. This figure is produced by modifying Kimura 2020 [73].

**Figure 7 biomolecules-14-00129-f007:**
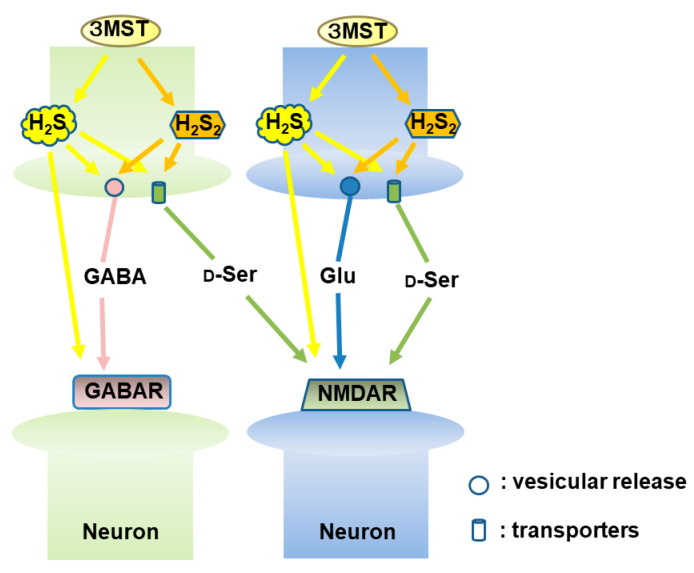
H_2_S and H_2_S_2_ induce the release of neurotransmitters. H_2_S more efficiently induces the release of GABA and glutamate than H_2_S_2_. Both H_2_S and H_2_S_2_ induce D-serine release with a similar potency. H_2_S/H_2_S_2_ may alter the synaptic vesicle dynamics and/or transporters to facilitate the release of neurotransmitters. This figure is produced by modifying Furuie et al. (2023) [41].

**Figure 8 biomolecules-14-00129-f008:**
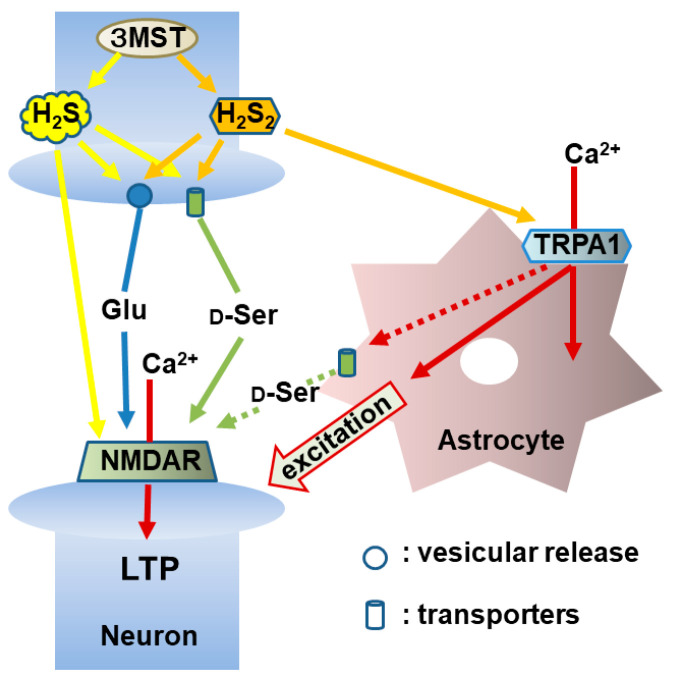
H_2_S and H_2_S_2_ facilitate the induction of hippocampal long-term potentiation (LTP), a synaptic model of memory formation. H_2_S and H_2_S_2_ induce the release of glutamate and D-serine to enhance the activity of NMDA receptors. H_2_S reduces the cysteine disulfide bond localized to the amino terminus of NMDA receptors, while H_2_S_2_ activate TRPA1 channels in astrocytes to increase the activity of nearby neurons to facilitate the induction of LTP. This figure is produced by modifying Furuie et al. (2023) [41].

**Figure 9 biomolecules-14-00129-f009:**
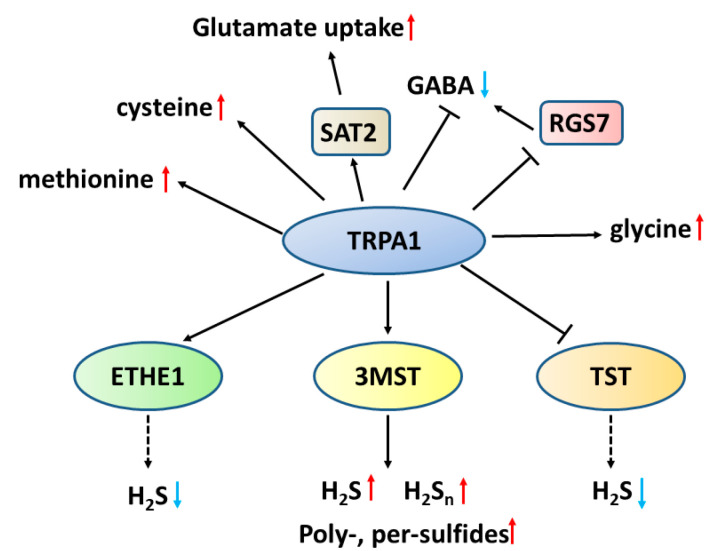
The involvement of TRPA1 channels in the metabolism of sulfur containing amino acids and other neurotransmitters. Deficiency of TRPA1 channels decreases the levels of methionine, cysteine, glycine, and 3MST, while increases those of GABA [40,41]. Suppression of TRPA1 channels by resiniferatoxin, an antagonist of TRPV1 and TRPA1 channels, upregulates spermidine/spermine N(1)-acetyltransferase 2 (SAT2), which regulate the levels of glutamate, while suppresses the regulator of G protein signalling 7 (RGS7), which regulates synaptic plasticity and GABA signalling [42].

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
