# Peer review of "Hydrogen Sulfide (H2S)/Polysulfides (H2Sn) Signalling and TRPA1 Channels Modification on Sulfur Metabolism"

_biomolecules, 2024, doi:10.3390/biom14010129_

Round 1

Reviewer 1 Report

Comments and Suggestions for Authors

The review ” Hydrogen sulfide (H2S)/polysulfides (H2Sn) signalling -Neurotransmitters release and 3MST interaction with TRPA1 channel” by H. Kimura describes the current evidence on the mechanisms of action of hydrogen sulfide/ polysulfides in the central nervous system, as well as its role in  physiological and pathological functions. Despite the nice and logic decription of H2S/H2Sn effects I noticed that Prof Kimura mainly refers to his own studies and to less extend considers the studies of other groups.

Minor comments

1.       The parts of review about mechanisms of H2S and H2Sn actions in neuronal transmission  and part 9 “The release of neurotoransmitters by H2S and H2Sn” mainly use data obtained by the author and do not include the possible targets of H2S in presynaptic terminals involved in transmitter release which was investigated in the central and peripheral nervous system.

I suggest to add to text  of manuscript more information about possible mechanisms of H2S/H2Sn effects which may include presynaptic ion channels and receptors, ryanodine receptors, the proteins of exo- and endocytosis (See Yi Luo  et all .,  CNS Neurosci Ther, 20: 411-419. https://doi.org/10.1111/cns.12228. Rodkin et al ., International Journal of Molecular Sciences 2023, 10.3390/ijms241310742, 24, 13, (10742). Paul BD, Snyder SH. Antioxid Redox Signal. 2015 Feb 10;22(5):411-23. doi: 10.1089/ars.2014.5917,1.            Kumar M, Sandhir R. CNS Neurol Disord Drug Targets. 2018;17(9):654-670. doi: 10.2174/1871527317666180605072018. PMID: 29866024.,   Gerasimova E, Lebedeva J, Yakovlev A, Zefirov A, Giniatullin R, Sitdikova G. Mechanisms of hydrogen sulfide (H2S) action on synaptic transmission at the mouse neuromuscular junction // Neuroscience. 2015 Sep 10;303:577-85. doi: 10.1016/j.neuroscience.2015.07.036; Mitrukhina, O.B., Yakovlev, A.V. & Sitdikova, G.F. The effects of hydrogen sulfide on the processes of exo- and endocytosis of synaptic vesicles in the mouse motor nerve endings. Biochem. Moscow Suppl. Ser. A 7, 170–173 (2013). https://doi.org/10.1134/S1990747812050121)

 2.       The author described that H2S/H2Sn produced only astrocytes but there are several data indicate that CBS expressed in neurons  (See Robert K, Vialard F, Thiery E, et al. Expression of the Cystathionine β Synthase (CBS) Gene During Mouse Development and Immunolocalization in Adult Brain. Journal of Histochemistry & Cytochemistry. 2003;51(3):363-371. doi:10.1177/002215540305100311)

3.      From the the text and in Figure 2 and 4  the mechanism of facilitating effect of H2S/H2Sn  is not clear, what substance releases from astrocytes which involves in in LTP (Fig 2) and what mechanisms induces the release of D serin from presynaptic terminal (Fig 4)?

4. In the part.10 the author  described  resiniferatoxin as  an ultrapotent capsaicin analogue agonist of TRP vanilloid type 1 (TRPV1) and TRPA1 channels (line 453) or antagonist (line 467)?? However the refered paper points that  resiniferatoxin is a bloker of TRPV1 receptor only. At the same time H2S was also shown as an agonist of TRPV1 (Koroleva et al Front. Cell. Neurosci 2017. 11:226. doi: 10.3389/fncel.2017.00226 ).  Therefore this hypothesis cannot explain the connection between TRPA1 activation and sulfut metabolism.  Additionally (line 479), how can TRPA1 regulate the activity of CAT, when TRPA1 is active only in astrocytes?

5 Line 423 - wrong citation

Author Response

Thank you for your comments and suggestions. The followings are answers to the comments.

  1. The previously reported possible mechanisms for the release of neurotransmitters induced by H2S/H2Sn were added to the discussion in the revised manuscript (lines 379-385)

Mitrukhina et al. reported that NaHS increased the frequency of miniature end-plate potentials and the amplitude of the evoked postsynaptic responses to a single stimulation in the motor nerve ending of the mouse diaphragm, suggesting the enhanced synaptic vesicle exocytosis and reduced endocytosis [132]. The same group showed that a similar observation was obtained in the neuromuscular junction showing the involvement of intracellular Ca2+, cAMP and presynaptic ryanodine receptors in the vesicular release of neurotransmitters [133].

  1. The localization of CBS was discussed (lines 124-127).

CBS was reported to be also localized to neurons using the polyclonal antibody preincubated with CBS knockout mouse brain extract to mask any nonspecific immunoreactivity [55]. However, not only our antibody but slso the same antibody of Robert et al. after affinity purification detected CBS in astrocytes but not in neurons [15].

  1. The possible mechanisms for the release of neurotransmitters by H2S/H2Sn were added to Fig. 7 (original Fig. 4) legend and text (408-415).

H2S/H2S2 may alter the synaptic vesicle dynamics and/or transporters to facilitate the release of neurotransmitters.

The release of D-serine by H2S2 from the hippocampus of TRPA1 knockout rats showed no significant difference with that in the wild-type rats [41]. However, LTP was not induced in TRPA1 knockout rats that agree with the observation by Shigetomi et al. [127]. The synaptic transmission is regulated by the interaction between neurons and astrocytes, and astrocytes-dependent facilitation regulates the vesicular release of neurotranamitters including glutamate in the hippocampal synapses (Fig. 8) [135]. These observations suggest that the activation of TRPA1 channels in astrocytes is required for the induction of LTP but that the channels are not involved in the release of D-serine [41].

  1. It was reported that resiniferatoxin suppresses both TRPV1 and TRPA1 channels.  Pecze L, Pelsoczi P, Kecskés M, Winter Z, Papp A, Kaszás K, Letoha T, Vizler C, Oláh Z. Resiniferatoxin mediated ablation of TRPV1+ neurons removes TRPA1 as well. Can J Neurol Sci. 2009, 36, 234-41. 

(Original line 479, present line 482). The interaction between astrocytes and neurons on the transport of cysteine was added in the revised manuscript.

5. Citation was corrected.

Reviewer 2 Report

Comments and Suggestions for Authors

The manuscript by Hideo Kimura, entitled "Hydrogen sulfide (H2S)/polysulfides (H2Sn) signalling -Neurotransmitters release and 3MST interaction with TRPA1 channels-" opens new fields of investigation about the role of H2S and H2Sn in CNS's diseases. In my opinion the manuscript only requires minor revisions (Listed below) but can be improved by a graphical point of view. Indeed the biochemical processes described in paragraph 4, 5, 6, 7 can be more clear if some figures with descriptive captions will be added.

Minor revisions (typos):

Row 64: repored should be changed in reported

Row 131 postsynapase should be changed in postsynapse

Row 297 What is "thiyl"?

Row 303 peritonial should be changed in peritoneal

Row 304 part of the sentence should be changed in "...of the H2S donor GYY4137 and of the NO donor DEA NONOate.."

Row 437 licalized should be changed in localized

Row 492-493 the sentence is incomplete

Row 556 neurotranamitters should be changed in neurotransmitters

Row 565-566 the references are reported both by numbers and by authors' surnames

Comments on the Quality of English Language

the quality is sufficient to understand the topic. 

Author Response

Thank you for your comments and suggestions.

Figures with descriptive captions were added to the revised manuscript.

Typos were corrected as suggested. Thiyl radical (HS.) was shown.

Reviewer 3 Report

Comments and Suggestions for Authors

This manuscript summarizes the metabolism, biological functions, and molecular mechanisms of the gasotransmitter H2S and its key enzyme; polysulfides, as well as the main enzyme 3MST involved in their production in the central nervous system. It also elucidates the interaction between H2S, polysulfides, and NO, their regulation of neurotransmitters, and the molecular mechanisms regulating  protein post-translational modifications, including TRPA1.
Questions:
1. There is considerable repetition in the content throughout the text.
2. Although the author describes the regulation of TRPA1 in various sections, there is no focused discussion on its regulatory mechanism, which does not align with the title. It is suggested to modify the title.
3. The overall logical structure lacks detail and needs reorganization. For instance, "2. Identification of H2S in the brain" could be combined with "4. Production and metabolism of H2S" and "5. Production of H2Sn and other S-sulfurated molecules." Combine the content of "6. 7. 8." with appropriate subheadings for clarity.
4. TRPA1 can be separately addressed as a distinct section.

Comments on the Quality of English Language

None.

Author Response

Thank you for your comments and suggestions.

  1. The repetition was deleted.
  2. The title was modified as Hydrogen sulfide (H2S)/polysulfides (H2Sn) signalling

-Neurotransmitters release and TRPA1 channels on sulfur metabolism-

  1. The overall logical structure was reorganized.
  2. The effect of TRPA1 on sulfur metabolism was separately addressed.

Round 2

Reviewer 3 Report

Comments and Suggestions for Authors

There is no any question.